# Ultrafast energy quenching mechanism of LHCSR3-dependent photoprotection in *Chlamydomonas*

Mengyuan Zheng [1,2,3,4], Xiaojie Pang [1,2,3,4], Ming Chen[1,2] & Lijin Tian [1,2,3] ✉

Photosynthetic organisms have evolved an essential energy-dependent quenching (qE) mechanism to avoid any lethal damages caused by high light. While the triggering mechanism of qE has been well addressed, candidates for quenchers are often debated. This lack of understanding is because of the tremendous difficulty in measuring intact cells using transient absorption techniques. Here, we have conducted femtosecond pump-probe measurements to characterize this photophysical reaction using micro-sized cell fractions of the green alga *Chlamydomonas reinhardtii* that retain physiological qE function. Combined with kinetic modeling, we have demonstrated the presence of an ultrafast excitation energy transfer (EET) pathway from Chlorophyll a (Chl a) $Q_y$ to a carotenoid (car) $S_1$ state, therefore proposing that this carotenoid, likely lutein1, is the quencher. This work has provided an easy-to-prepare qE active thylakoid membrane system for advanced spectroscopic studies and demonstrated that the energy dissipation pathway of qE is evolutionarily conserved from green algae to land plants.

Microalgae are considered a promising candidate for both biofuel production and large-scale recapture of emitted $CO_2$[1,2]. However, their application is severely hindered by the low economic viability[2,3]. One of the bottlenecks is the ultra-low efficiency of sunlight energy utilization compared to commercial inorganic solar panels[1,3]. Nevertheless, the algae-based approach still holds high potential, especially considering that the massive efforts in improving solar cells' efficiency over the past decades have never been given to a bio-solar system. The most significant energy loss occurs in photosynthetic light harvesting, the initial step of photosynthesis[1,2], especially in a so-called qE process. This heat dissipation mechanism protects plants against photodamage from high light[4], however, the energy loss in qE is so significant that nearly 80% of the total absorbed energy on a single photosystem II complex level is wasted[5,6]. Recently, genetic engineering approaches have been taken and approved to optimize the qE process and thus avoid unnecessary excitonic energy loss to be practical for higher plants and algae[7–9]. Despite all the progress in biology regarding qE engineering, the physics underlying this energy dissipation

mechanism still needs to be clarified, thereby limiting further improvement.

Photosystem I and II (PSI and PSII) supercomplexes are two major photocatalysts in the photoreactive phase of photosynthesis[10], in which the hazardous reactive oxygen species (ROS) are mainly produced[11]. Therefore, both are, in principle, the targets of photodamage and thus require protection by qE, directly or indirectly[4,12–14]. In general, most studies considered that qE is predominantly associated with PSII rather than PSI since the latter itself has such a rapid fluorescence decay (less than 100 picoseconds) that its fluorescence quenching effect, if present, is less visible compared to PSII[4,15–18]. On the other hand, PSI also needs photoprotection; its inhibition originates more from an unbalanced electron flow[19,20]. Thus, reports point out that the qE protection of PSII alleviates pressure on the electron donor side of PSI and concomitantly protects PSI[13,14]. More details on the qE mechanism, its physiological roles, and its evolutionary diversity can be found in refs. 4,18,21,22.

In the green alga *Chlamydomonas reinhardtii*, nonphotochemical quenching (NPQ) activity requires the Light Harvesting Complex

[1]Key Laboratory of Photobiology, Institute of Botany, Chinese Academy of Sciences, Beijing 100093, China. [2]China National Botanical Garden, Beijing 100093, China. [3]University of Chinese Academy of Sciences, Beijing 100049, China. [4]These authors contributed equally: Mengyuan Zheng and Xiaojie Pang. ✉e-mail: ltian@ibcas.ac.cn

Stress-Related proteins (LHCSRs)[23,24], which belong to the LI818 family[24-29]. Unlike PsbS, which is involved in qE in land plants[30,31], LHCSRs are pigment-binding proteins[26], qualifying them as potential energy quenchers. Both LHCSRs are pH sensors that sense the low luminal pH, activating the qE type of energy quenching of the PSII supercomplexes[32] and might also quench the PSI by quenching the associated LHCII antenna[24]. Structurally, it has already been determined that the LHCSR in dimer binds at three positions on the PSII antenna $(C_2S_2)$[33], yet an atomic-resolution structure of PSII-LHCSR3 super-complex remains lacking. Regarding the quenching kinetics of qE, only optical properties of the reconstituted LHCSR1[34-36] and LHCSR3[37-39] proteins have been assessed, but how exactly a native LHCSR protein interacts with and quenches the complexes of PSII is still one of the challenging questions in photosynthesis research.

Ultrafast transient absorption (TA) spectroscopy allows the investigation of many biological processes that occur on very fast time scales from femtoseconds to nanoseconds[40]. It has often been used to identify energy decay kinetics in natural light-harvesting systems, including energy quenching observed in LHCII (the primary photosynthetic antenna in higher plants) aggregates[41-44]. However, the application of TA on intact cells is hardly successful because of the high scattering of intact cells and difficulties in maintaining the cells in a stable qE state through the measurements (usually lasting for hours)[45,46]. Indeed, an in vitro system of qE with minimized light scattering would facilitate spectroscopic studies. However, such a system is difficult to reconstruct since membrane protein functions hardly survive harsh solubilizations by detergents. In recent years, a promising platform of nanodiscs that could circumvent these challenges has been extensively explored[47-51]. It is based on a detergent-free method to keep best possible the membrane proteins in their near-native environment and maintain their functions. The small size of the nanodiscs favors advanced spectroscopic studies. Nevertheless, a nanodisc system mimicking the LHCSR-related qE process is not yet available.

Herein, we took another approach to overcome these technical issues, as discussed above, other than nanodiscs. We prepared the micro-sized cell lysates of *C. reinhardtii* that are low in scattering and, more importantly, still carry its biologically active qE for spectroscopic studies. Using these samples, we characterized the quenching mechanism of qE by performing both steady-state fluorescence and femtosecond TA spectroscopy.

## Results and discussion
### Sample preparation and characterization
To avoid the fluorescence variations caused by state transition, the other major form of NPQ in *Chlamydomonas* (accounting for 30–40% of NPQ)[52-56], but mainly focus on LHCSR3-dependent qE, we have studied the *C. reinhardtii* stt7-9 mutant that is deficient in state transition and used *npq4/stt7-9* double mutant that lacks both state transition and LHCSR3 mediated qE as a control. Supplementary Fig. 1 shows the preparation process of the micro-sized cell fractions, a prerequisite step for extracting many previously reported thylakoid membrane proteins. We found that although the concentration of chlorophylls decreased by 25% after disruption and centrifugation (red line in Supplementary Fig. 2A, B), the absorption peaks of pigments (435, 474, 652, and 676 nm) were unchanged before and after cell disruption (Supplementary Fig. 2C, D). This result indicates that most of the proteins on the thylakoid membrane are stably retained in our samples, further confirmed by the Western blot analysis results (Supplementary Fig. 3). Both samples contain the essential subunits of PSI/PSII complexes, LHCII, and additional LHCSR1. As expected, LHCSR3 was found in the *stt7-9* sample but not in *npq4/stt7-9*. To obtain more biochemical information about the samples, we also performed blue-native polyacrylamide gel electrophoresis (BN-PAGE), and the results showed that the PSI/PSII supercomplexes are both present in the micro-sized cell fractions and consistent between samples at two pHs

(Supplementary Fig. 4), a character that is critical for subsequent TA data analysis. Confocal fluorescence microscopy images of the samples show the size of thylakoid membrane fragments is between 1–3 μm² in surface area, and no significant differences were observed either between two mutants or between samples of the same mutant at two different pHs (Supplementary Fig. 5). This approach offered three advantages: i) the native environments of LHCSR3 are maintained, thus, its physiological functions are kept; ii) the high actinic light required to induce qE is replaced by lowering the solution pH in darkness[26,29]. In this way, the qE state can be sustained for hours in complete darkness, vastly simplifying the measurement; and more importantly, iii) the micro-sized cell fraction has greatly minimized optical scattering, making light probe in transmission mode possible (Supplementary Fig. 6). The optical density of the micro-sized cell fraction at 750 nm (mainly scattering) was only 0.08 (with path length 1 cm), which was much reduced from 1.55 for the intact cells with the same chlorophyll concentration, namely, the light transmission was $10^{(1.55-0.08)} = 29.5$ times stronger for the cell fractions than intact cells. Stronger light transmission maximizes the detection efficiency of transmitted probe light and concomitantly delivers high-quality data with a high signal/noise ratio, an essential prerequisite for further complex kinetics modeling.

### Steady-state absorption and fluorescence
Figure 1A, B show the steady-state absorption spectra of the micro-sized cell fractions of *stt7-9* and *npq4/stt7-9* at different pHs, respectively. No significant differences were observed between samples at different pHs, indicating that a lower pH condition does not affect the absorption coefficient of the samples and the observed fluorescence quenching (Fig. 1C) was a pure quenching effect on the Chl excited state. For the quenched state at pH 5.5, roughly 60% of the energy absorbed was dissipated compared to the unquenched state at pH 7.5 for the *stt7-9* mutant. In contrast, hardly any quenching effect was observed for the *npq4/stt7-9* double mutant (Fig. 1D), highlighting the involvement of LHCSR3 (Supplementary Fig. 3). These results are consistent with the steady-state chlorophyll fluorescence traces of the intact cells (Supplementary Fig. 7): both actinic light and the addition of HAc can induce fluorescence quenching in *stt7-9* cells, while the double mutant of *npq4/stt7-9* only showed a slight decrease due to the presence of a small amount of LHCSR1 protein[23]. Besides, we found that the fluorescence quenching levels were independent of the excitation wavelengths (Supplementary Fig. 8), indicating that PSIIs, the main contributor to steady-state fluorescence[57-60], stayed largely intact and fast energy equilibrium within these PSII complexes retained after the ultrasound sonication[61]. Further protein solubilization using detergents would inevitably lead to significant heterogeneities of LHC antennas and photosystem supercomplexes[62-64]. These results again showed the merit of keeping the membrane proteins in their native lipids environments. Even without artificial destructions, the system is already fairly complex, e.g., PSI complexes present, although with a small contribution to the steady-state fluorescence[58]. In addition, at pH 5.5, only part of the PSII supercomplexes was quenched by LHCSR3, thus causing heterogeneity of PSII in different fluorescence states, including both PSII_unquenched (PSII_UQ) and PSII_quenched (PSII_Q). These two states were also included in the data analysis of time-resolved absorption, see below.

### Transient absorption
To investigate the quenching mechanism of qE further, we characterized the femtosecond TA kinetics by comparing the quenched and unquenched states. The low-energy pigments in the samples were preferentially excited by using 675 nm pump pulses, and signals were probed in the visible and near-infrared wavelength range (500–1000 nm, Fig. 2 and Supplementary Fig. 9). Consistent with the steady-state fluorescence data, the Chl a bleaching signal at pH 5.5 decays faster than it at pH 7.5 (Supplementary Figs. 10A–C) for *stt7-9*

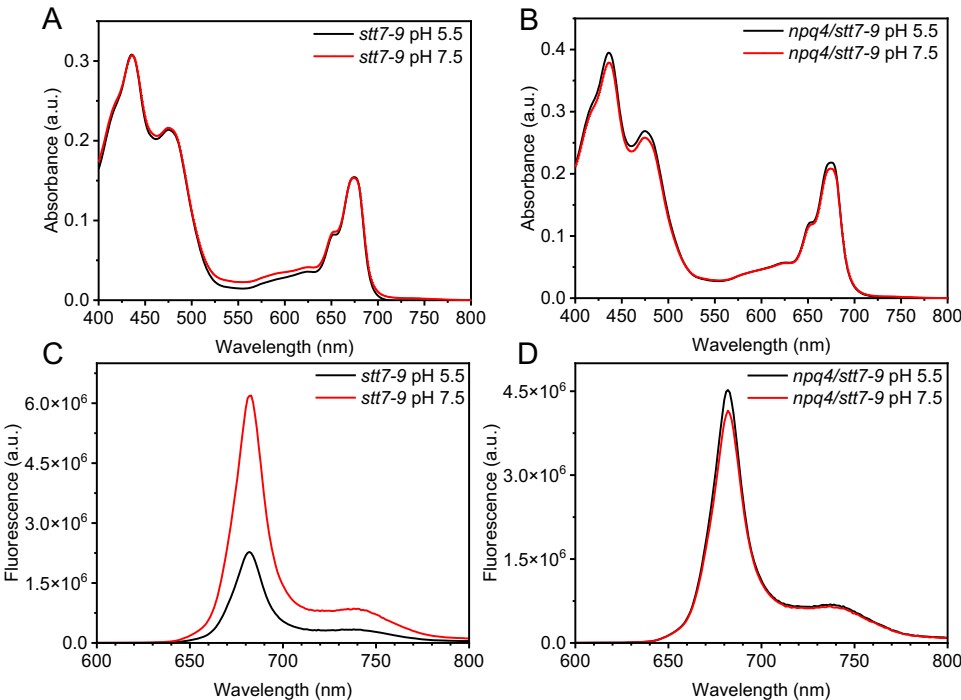

**Fig. 1 | Steady-state spectra of the micro-sized cell fractions of *C. reinhardtii* cells of *stt7-9* and *npq4/stt7-9*. A, B,** respectively represent the room temperature absorption spectra of *stt7-9* and *npq4/stt7-9* at both pH 5.5 and 7.5. Both were measured without using an integrating sphere. **C, D,** respectively represent the room temperature fluorescence spectra of *stt7-9* and *npq4/stt7-9* at pH 5.5 and 7.5. Excitation was made at 430 nm. Source data are provided as a Source Data file.

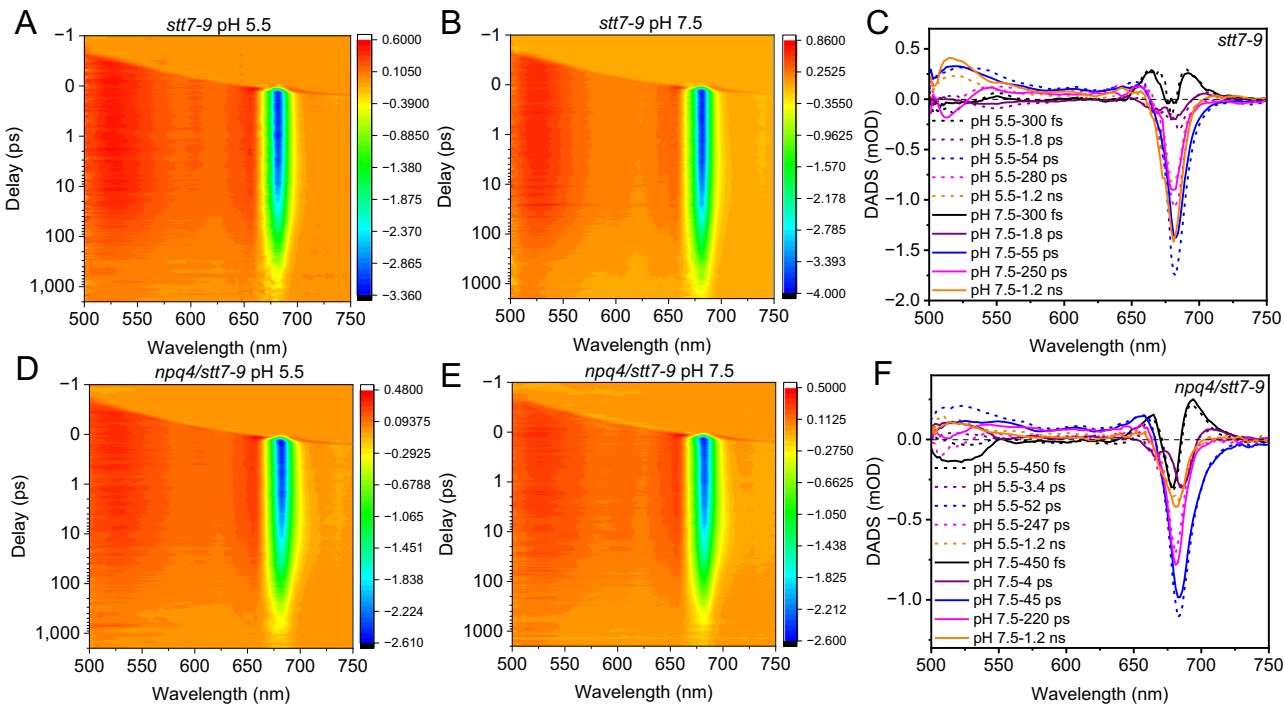

**Fig. 2 | Transient absorption (TA) data of *stt7-9* and *npq4/stt7-9* micro-sized cell fractions at pH 5.5 and 7.5.** TA images for *stt7-9* (**A, B**) and *npq4/stt7-9* (**D, E**) with excitation at 675 nm, and the DADSs were calculated for *stt7-9* (**C**) and *npq4/stt7-9* (**F**), spectra correspond to quenched sample at pH 5.5 (dotted line) and unquenched ones at pH 7.5 (solid line), respectively. The DADSs at different pHs were normalized to the total bleaching of Chl at around 680 nm at time zero. Source data are provided as a Source Data file.

sample but not for *npq4/stt7-9* mutant (Supplementary Fig. 10D–F). Global analysis was performed to extract quantitative information from the measured TA images (Fig. 2A, B, D, and E). To quantitatively describe the TA data, we have employed a parallel kinetics model of five compartments, which is a compromise between simplicity and sufficient fit, and we obtained the decay-associated difference spectra (DADS) and their corresponding lifetimes for *stt7-9* and *npq4/stt7-9* (Fig. 2C, F, and the fitting curves in Supplementary Figs. 12, 13), respectively. To

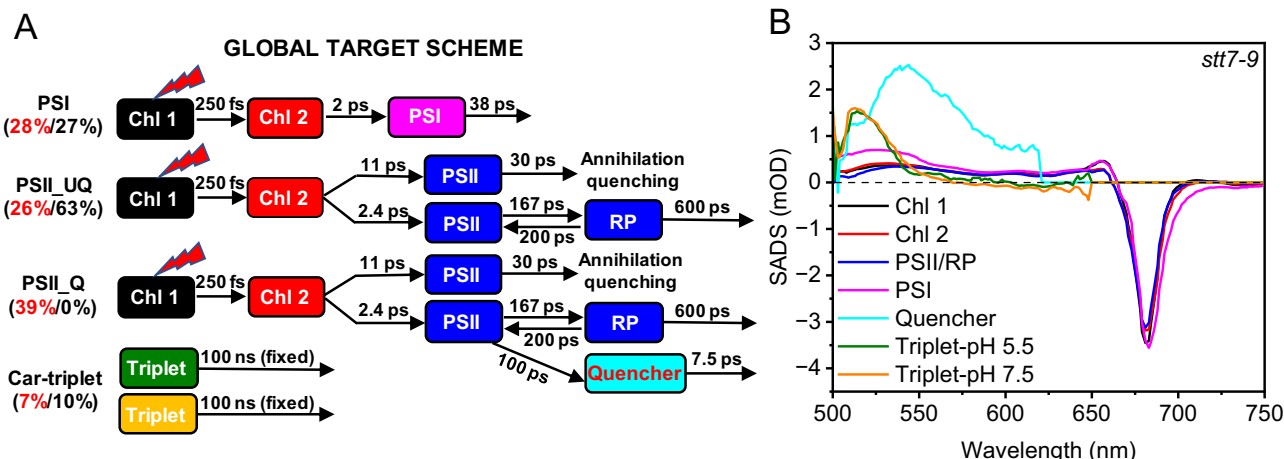

**Fig. 3 | Global target analysis of TA data of *stt7-9* micro-sized cell fractions at pH 5.5 and 7.5. A** The unified compartment target model for global target analysis of 675 nm excitation data. The spectra of the compartments between the data of different pHs have been linked. The percentages of red and black fonts represent the proportion of excitation energy of each branch in the model at pH 5.5 and 7.5, respectively. **B** SADS of each compartment were plotted: *stt7-9* micro-sized cell fractions at pH 5.5 and pH 7.5. Source data are provided as a Source Data file.

highlight the differences induced by quenching in the individual DADS curves, we also showed the individual DADSs, see Supplementary Fig. 11.

Energy equilibrium between different pools of chlorophylls (black and purple DADSs in Fig. 2C) takes place in 300 fs and 1.8 ps. These two processes were previously observed for the purified LHCs/aggregates, where excitation reaches equilibrium within sub-picosecond to a few picoseconds[42,43,65–67]. Note that there might be slower EET processes with lifetimes of around ten picoseconds in the system. However, our model does not resolve these because of its low population (due to the nearly complete spectral overlap between donor and acceptor molecules). Upon quenching, these two fast processes were both enhanced at 680 nm for the mutant of *stt7-9* (Supplementary Fig. 11A, B), but neither was changed in the double mutant of *npq4/stt7-9* (Supplementary Fig. 11F, G). These results indicate that the LHCSR-dependent quenching processes are fast enough to disturb this equilibrium.

The blue DADS with a lifetime of 54/55 ps respectively for pH 5.5 and 7.5 (blue DADS in Fig. 2C) reflect either a rapid energy quenching caused by LHCSR[29,68] or annihilation that happens in PSII[69,70] or photochemical quenching in PSI[71,72]. Upon quenching, this component was significantly increased in population from 37% to 48% for *stt7-9* (Fig. 2C and Supplementary Table 1), likely caused by LHCSR-induced PSII quenching. Note that annihilation or PSI was not considered responsible for this increase as both of them should not be affected by low pH in *stt7-9* at room temperature according to previous reports[24,29].

The remaining bleaching decays in 280/250 ps and 1.2 ns (magenta and orange DADS in Fig. 2C). These two lifetimes are typical for closed PSII reaction centers[68]. The 200 ps component was slightly more populated in quenching, whereas the 1 ns component decreased from 38% to 23% in amplitude compared to its corresponding one in the unquenched sample (Fig. 2C and Supplementary Table 1). The 1 ns component partially left after quenching suggests that a fraction of PSIIs were not quenched. A possible scenario here is that these two components of PSII_UQ, namely, 250 ps and 1.2 ns, were shortened to 50 ps and 280 ps, respectively, upon quenching. This synchronized lifetime shortening would explain why this 200 ps component increased slightly in population when reacting to quench since that part of the 1.2 ns component fell into this lifetime after quenching.

To quantify the quenching effect observed in the TA data, we reconstructed the steady-state fluorescence based on DADSs, see Supplementary Fig. 14. The total fluorescence decreased by roughly 40% at lower pH for the *stt7-9* mutant while changing marginally in the double mutant of *npq4/stt7-9*. The quenching level is lower than that observed 60% in the steady-state measurements (Fig. 1C). This

discrepancy is caused by the unavoidable multi-exciton annihilation effect in ultra-short laser spectroscopy, which accelerates the overall decay of the system and thus diminishes the quenching.

To summarize, all the lifetime components, their possible assignments, and how the proportions of the Chl-$Q_y$ bleaching at 680 nm to the total reaction to qE at this wavelength were summarized in Supplementary Table 1. As expected, for double mutant, the overall quenching effect on the DADSs (Fig. 2F) caused by LHCSR1 showed a similar pattern but much weaker as compared to the quenching in *stt7-9*, in which both LHCSR3 and LHCSR1 are functional.

## Target modeling of the transient absorption data

As the global analysis results showed above, qE influenced all the lifetime components. However, how fast the quenching was and what the direct quencher is need to be clarified. To address these questions, we conducted a global target analysis by simultaneously fitting the data of unquenched and quenched samples using a unified compartment target model (Fig. 3A and the fitting curves in Supplementary Fig. 15). In the model (Fig. 3A), each compartment represents a pool of pigments with their species-associated difference spectra (SADS) deconvoluted[73], and the compartments are energetically connected through energy transfer processes with varied energy transfer rates (shown in migration time). To account for the heterogeneity of photosystem supercomplexes and their different quenching states for the *stt7-9* samples, the model employs three energy transfer chains in a parallel scheme, namely, PSI, represented by three compartments, PSII_UQ and PSII_Q. Each branch of PSII contains five compartments, as suggested by the global analysis, and a fourth branch of carotenoids triplet, of which the presence of a long decaying lifetime (>ns) is visible from the raw datasets (Fig. 3A). The proportions of excitation energy on each branch were roughly estimated by equalizing the minima of the $Q_y$ bleaching in SADSs. Besides, annihilation must be included for each branch of PSIIs and an additional energy quenching pathway (qE) for PSII_Q. Note that compartments in the same colors were forced to have identical spectra to greatly simplify the model, whereas the spectrum of the quencher was kept free. To stabilize the fitting, several constraints were applied, as listed in Supplementary Text 1.

For the branch of PSI, after those two fastest energy transfer steps (black and red compartments in Fig. 3A), excitation was then relaxed within 50 ps (magenta compartment in Fig. 3A), which agrees with previous reports[71,72]. In our model, PSI roughly accounts for 30% of the total excitation. The spectrum of the PSI component (magenta SADS in Fig. 3B) has a visible shoulder between 690 and 720 nm compared to

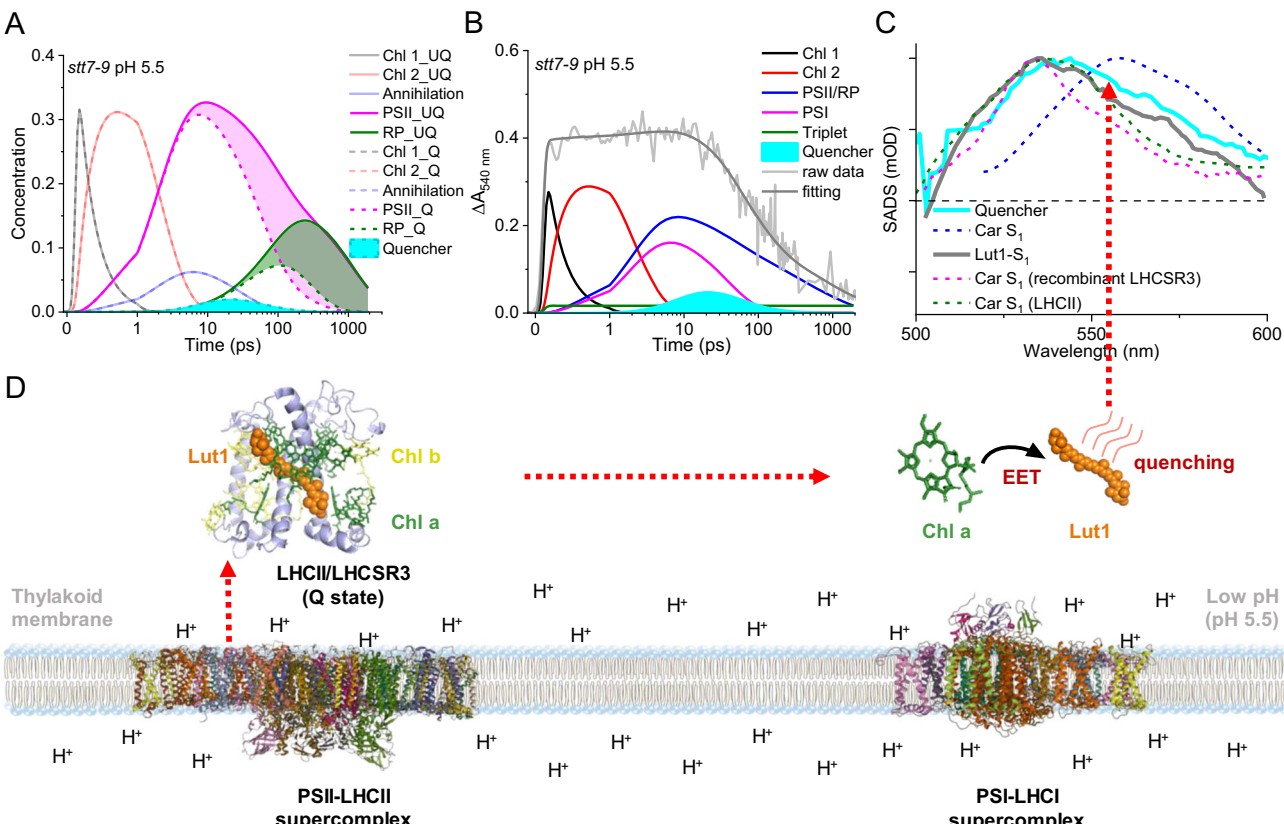

**Fig. 4 | Proposed energy quenching mechanism of LHCSR3-related qE in the native thylakoid membrane. A** Concentrations of the different species were estimated in the global target analysis (data of the *stt7-9* sample at pH 5.5 upon 675 nm excitation). For comparison, the solid curves are 1.5-fold magnified, and the areas in magenta and green represent the population changes of PSII and RP, respectively, due to quenching. For simplicity, the concentrations of two branches, PSI and Car-Triplet, are omitted. **B** The kinetics trace of the *stt7-9* pH 5.5 sample at 540 nm and the populations of different compartments at this wavelength. **C** The

spectrum of the quencher (solid cyan line) that overlaid with the ESA spectrum of carotenoid $S_1$ (blue dashed line, see details in Supplementary Fig. 19B) from TA data of *stt7-9* micro-sized cell fractions at pH 5.5 and several published ESA spectra of Lut1-$S_1$ state of LHCII aggregates (solid gray line)[43], carotenoid $S_1$ state of recombinant LHCSR3[38] (magenta dashed line) and carotenoid $S_1$ state of LHCII[76] (green dashed line). **D** Schematic of PSI and PSII supercomplexes together with LHCSR3 in the thylakoid membrane. Source data are provided as a Source Data file.

the other four components, which validates our assignment. Regarding the quenching, Girolomoni et al. reported that PSI is quenched in the mutant of *stt7-9* by LHCSR3 at 77K[24], but Tian et al. disagree with this opinion[29]. Despite this controversy, PSI quenching at room temperature is certainly negligible because of the fast kinetics of PSI[74]. Therefore, no quenching pathway was given for PSI at room temperature.

For the branches of PSII, after energy equilibrium is reached between the first two chlorophyll pools (black and red compartments in Fig. 3A), excitons further decay into the bulk chlorophyll compartment of PSII (blue compartment in Fig. 3A) through two sub-pathways that respectively lead to singlet-singlet annihilations and the PSII charge separation. The former pathway dissipated roughly 18% of the excitation in 30 ps, while the second one relaxed the other 82%. Unfortunately, with our setup, an annihilation-free condition could not be reached. Excitons in PSII (blue compartment in Fig. 3A) generate the radical pair state (RP) that further decays in 600 ps. The quenching pathway was added to the PSII (blue compartment in Fig. 3A), which was depicted by a typical "inverted kinetic model" that started with a slower energy transfer from PSII to the quencher (cyan compartment in Fig. 3A) in 100 ps and thereby a faster quenching in 7.5 ps. At pH 5.5, PSII_Q absorbs 39% of the total excitation and PSII_UQ absorbs 26%. Therefore, the quenched fraction accounted for ~60% of the total energy absorbed by PSII. At pH 7.5, PSIIs are in their fully unquenched state. All the obtained SADSs, including the first two Chl pools, PSII and RP (black, red, and blue SADS in Fig. 3B), have Chl $Q_y$ bleaching in a similar line shape and amplitude.

For the branch of carotenoids triplet state, the proportion of excitation energy inputs on the triplet was at variance for quenched and unquenched samples (7% at pH 5.5 and 10% at pH 7.5). The triplet spectra (green and orange SADS in Fig. 3B) have a peak around 510 nm, of which the lifetime was fixed at 100 ns in the fittings.

This model also fits the double mutant of *npq4/stt7-9* well after removing the branches of PSII_Q (no LHCSR3) and Car-triplet, see the fitting results in Supplementary Fig. 16. The absence of a Car-triplet in the raw data may be because the effective exciting intensity on PSII is lower since the mutant contains higher proportion PSI (~60%), which could shade PSII, thus producing less triplet state. The fitting of the double mutant was performed independently and resulted in a high-quality fit with realistic SADS (Supplementary Fig. 17). Therefore, it can be considered an unbiased validation of our model.

From our model in Fig. 3, the photoproduct formed by the quencher molecule peaks around 540 nm, and its population is very small, as shown in Fig. 4A, B. Despite the signal being low, we could still notice its presence in the raw data, namely, the initial excited-state absorption (ESA) signal (<50 ps) is stronger for the quenched sample at pH 5.5 (Supplementary Fig. 18A–C) but weaker at ns because of the quenching. The spectrum of ΔESA signal between quenched and unquenched samples largely resembles the spectrum of quencher that resolved in target analysis (Supplementary Fig. 18D), eliminating the possibility that the latter one is a fitting artifact. To better elucidate the nature of the quencher, we have compared the SADS of the quencher obtained here to the ESA of Car $S_1$ state of different systems from

published data (Fig. 4C), including the ESA spectrum of Car $S_1$ state of our *stt7-9* sample. We found that the ESA spectrum (cyan line in Fig. 3B and cyan SADS in Fig. 4C) of the quencher is very similar to the one of the Lut1-$S_1$ state (gray line in Fig. 4C) that resolved previously in the heavily quenched LHCII aggregates[43]. Therefore, we tentatively assigned the quencher molecule to lutein, possibly the one that binds at the Lut1 position[75]. Note that this photoproduct formed during qE differs from the Car $S_1$ spectrum (blue line in Fig. 4C) obtained upon direct carotenoid excitation (see experimental details in Supplementary Fig. 19). The latter peaks at 556 nm (blue line in Fig. 4C), -16 nm red-shifted from the quencher (cyan line in Fig. 4C) and the Car $S_1$ state of LHCII (green line in Fig. 4C)[76]. This red shift might be caused by a large amount of β-carotene in photosystems, whose $S_1$ excited state absorbs around 560 nm as reported previously[77,78]. Could the Car $S_1$ state observed here be an artificial electronic species introduced by strong laser pulses[77]? This possibility was ruled out in our global target analysis, in which we fitted the unquenched and quenched samples simultaneously, and only the differences related to qE were targeted. Another critical question is whether this quencher belongs to LHCSR3 or its interaction partners of LHCs in PSII[79]. Since the spectrum of the quencher is also similar to the ESA of Car $S_1$ state that reported for the recombinant LHCSR3 (magenta line in Fig. 4C), which might originate from Lut1 too[38], we thus could not determine the exact location of this lutein only based on its spectral signature.

Finally, to date, mainly four quenching mechanisms have been proposed to interpret the quenching for different LHC proteins: i) energy transfer from Chl to carotenoid[32,37,38,42–44,46,80]; ii) Chl-carotenoid excitonic coupling[81]; iii) Chl-carotenoid charge transfer[82]; and iv) Chl-Chl charge transfer without the involvement of carotenoid[83–85]. The former two pathways belong to the excitation energy transfer (EET) processes, while the latter two are essentially charge-transfer processes. Here, we failed to detect any absorption changes that originate from Car cations in the near-infrared region (Supplementary Fig. 9), thus excluding the model of Chl-carotenoid charge separation[26,36,37,86]. We demonstrated with our model that a carotenoid $S_1$ state was excited during the quenching. Hence, the process of Chl-Chl charge transfer less likely plays a role here, although we could not entirely exclude its involvement. We propose that the EET from Chl a to carotenoid is responsible for the LHCSR3-dependent qE process. Note that the PSII supercomplexes (-200 pigments) were quenched in 100 ps as reported previously and in this work[29], meaning that the quenching rate per molecule must be above $(1\,ps)^{-1}$ even if we assume that energy equilibration within PSII is instantaneous. The weak Coulomb-mediated Chl-car interaction when a dipole-dipole approximation was employed could not explain the high rate of energy quenching since the $S_0$ to $S_1$ state transition in the carotenoid is optically forbidden unless the $S_1$ state carries part of the $S_2$ oscillator strength due to symmetry-breaking of the carotenoid, as reported previously[87]. Alternatively, the strong Coulomb interaction and/or the exchange term in the Chl-car coupling dominates the energy transfer process[88,89]. This speculation agrees with the recent work of Ruan et al.[47], who assigned the energy transfer process from Chl to Lut to short-range (Dexter) energy transfer. They observed a sharp decline in lifetimes (from ns to a few hundred picoseconds per LHCII trimer) when the distance between Chl and Lut becomes closer than 5.6 Å in the purified LHCII in nanodiscs. This steep distance dependence could be fitted well with a Dexter-type EET model. Notably, compared to the LHCII in nanodiscs, the quenching process in a native membrane, as presented here, is happening at least ten times faster (-100 ps per PSII complex).

Our results demonstrate that the LHCSR3-related qE in native thylakoid membrane was realized via an ultrafast EET pathway from chlorophyll a (Chl a) $Q_y$ to carotenoid (possibly Lut1) $S_1$ state. This result supports the Chl→Car EET model that has been proposed for the qE mechanism in *Nannochloropsis oceanica* and higher plants, highlighting the conservation of this pattern of energy quenching from algae to higher plants regardless of the pivotal differences in the key qE players.

## Methods

### Cell growth and sample preparation

Cells of the *stt7-9* mutant were grown under high light (350 μmol photons·m$^{-2}$·s$^{-1}$), and the *npq4/stt7-9* mutant were grown under low light (50 μmol photons·m$^{-2}$·s$^{-1}$) in high-salt medium at 25 °C. Both the *stt7-9* and *npq4/stt7-9* mutants were gifts from Dr. Roberta Croce (Vrije Universiteit Amsterdam). All the materials used in the study are available from the corresponding author upon request. The cells grown in the logarithmic growth phase were collected by centrifugation ($1100 \times g$) for 4 min. Then, they were resuspended with lysis buffer (5 mM HEPES, 1 mM MgCl$_2$, 1.5 mM NaCl, and pH adjusted to 5.5 by adding acetic acid to keep the samples in a quenched state). An ultrasonic homogenizer (10200321, SCIENTZ-IID) was used to disrupt the cells. The ultrasonic power was set to 60 W, and the resuspended cells were lysed by sonication in an ice bath for 15 minutes (1 s sonication every 5 s). To remove the solid phase, samples were centrifuged at $20,000 \times g$ for 5 min at 4 °C, and the supernatant was used for subsequent experiments. To fully relax the quenching, KOH was added to titrate the pH of the medium back to 7.5, and to ensure all the PSII RCs closed, 50 μM dichlorophenyldimethylurea was added and pre-illuminated by weak light for 1 min before measurements.

### Steady-state spectra measurements

Absorption spectra were collected at room temperature using a UV−VIS Spectrophotometer (Cary 4000 UV−Vis, Agilent) equipped with an integrating sphere (DRA-900, Agilent). A cuvette with a path length of 1 cm (1 cm*1 cm*3 cm) was used, and the absorption spectra were recorded in the 400−750 nm range.

The room temperature steady-state fluorescence spectra were recorded using a spectrometer (FLS1000, Edinburgh Instruments). Three different excitation wavelengths of 430, 510, and 630 nm were used; a bandwidth of 3 nm was used for both excitation and emission.

Chlorophyll fluorescence traces were obtained with a pulse amplitude-modulated fluorimeter (Dual-PAM 100, Walz). All the saturating light (6000 μmol photons·m$^{-2}$·s$^{-1}$, 250 ms), actinic light (1500 μmol photons·m$^{-2}$·s$^{-1}$), and measuring light used in measurements were peaking at 630 nm. Before measuring qE, cells were dark-adapted for 10 min, and qE was activated under the illumination of actinic light (1500 μmol photons·m$^{-2}$·s$^{-1}$) for 150 s. A complete qE relaxation in darkness followed.

### SDS polyacrylamide gel electrophoresis (SDS-PAGE) and Western blot

The micro-sized cell fractions were separated by SDS-PAGE (12%), and immunoblots were conducted as described[23]. Protein was quantified by using the bicinchoninic acid (BCA) assay[90]. Antibodies against LHCSR3 (AS142766, Agrisera), LHCSR1 (AS142819, Agrisera), and Psab (AS10695, Agrisera) were used at 1:5000 dilution. The D1 antibody was a gift from Guangye Han (Institute of Botany, Chinese Academy of Sciences), and the Lhcb1 antibody from Zhenfeng Liu (Institute of Biophysics, Chinese Academy of Sciences). Both antibodies were used at 1:1000 dilution. The blots were imaged using a chemiluminescence imaging system (BLY-9650, BOLAIYAN).

### BN-PAGE analysis

BN-PAGE was performed with a 4% stacking, and 4−15% gradient resolving gel (4561085, BIO-RAD), and the gel was run at 4 °C with increasing voltage[91]. The micro-sized cell fraction samples were prepared at pH 5.5, as described above, and the samples were adjusted to pH 7.5 before solubilization. A total of 6 μg of Chl was loaded per lane, and the final detergent concentration was 1.5% β-DM.

## Confocal microscopy experiments

A minimal quantity (5 µL) of samples was pipetted onto a clean glass slide and sealed with a clean glass coverslip. The slides were visualized using a Zeiss LSM 980 inverted fluorescence microscope under a 100× oil immersion objective. The excitation wavelength was 488 nm, and the fluorescence signal was observed between 650–750 nm. Images were acquired with the same magnifications.

## Femtosecond transient absorption spectroscopy measurements

Pump-probe measurements were performed at room temperature with a HARPIA-TA ultrafast TA spectrometer system. The PHAROS/CARBIDE laser (PH2-UP, Light Conversion) delivers a 25 kHz 1030 nm pulse train, part of it drives the 100 fs visible non-collinear optical parametric amplifier (NOPA, ORPHEUS-N-2H, Light Conversion), which is used as a selective excitation light source. The other part of the 1030 nm light generates a white light supercontinuum, which serves as a probe. The samples were excited by 675 and 510 nm laser pulses with pulse energies of 3 and 5 nJ, respectively. TA spectra between 490–800 nm were recorded, and for 675 nm excitation, another spectral window in the 610–1000 nm region was additionally measured. The pump (~150 µm in diameter) and probe pulses overlapped at the sample at the magic-angle polarization. Samples were measured at an OD (670 nm) of 0.5 in a quartz cuvette with a path length of 2 mm (2 mm*1 cm*3 cm) and the samples were continuously stirred in the cuvette to prevent laser-caused damage during measurement.

## Global and target analysis

Global and target analysis was performed on the TA data with the R package TIMP-based Glotaran[73,92]. The application of a parallel model allowed us to estimate a set of evolution-associated difference spectra and decay-associated difference spectra (DADS) and their corresponding lifetimes capable of describing the temporal evolution of the data. Target analysis was used to extract the difference spectra SADS of the actual species involved in the measured excited-state kinetics. See a detailed description of modeling in Supplementary Text 1.

## Statistics and reproducibility

All the experiments were performed in triplicate. No statistical method was used to predetermine sample size. No data were excluded from the analyses. The experiments were randomized and the investigators were not blinded to allocation during experiments and outcome assessment.

## Reporting summary

Further information on research design is available in the Nature Portfolio Reporting Summary linked to this article.

## Data availability

All data are available in the article and its Supplementary files. The data that support the findings of this study are available from the corresponding author upon request. Source data are provided with this paper.

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

## Acknowledgements

We thank Dr. Roberta Croce (Vrije Universiteit Amsterdam) for her inspiration and support in the early stages of this project. We thank Dr. Jingyi Zhu (Dalian Institute of Chemical Physics, Chinese Academy of Sciences) for valuable discussions. Also thank Dr. Zhenfeng Liu (Institute of Biophysics, Chinese Academy of Sciences) for providing the Lhcb1 antibody and Dr. Guangye Han (Institute of Botany, Chinese Academy of Sciences) for the D1 antibody. This work is supported by grants from the National Key R&D Program of China (2019YFA0904600 and 2022YFC3401803 to L.T.), Science & Technology Specific Projects in Agricultural High-tech Industrial Demonstration Area of the Yellow River Delta (2022SZX12 to L.T.), and the National Natural Science Foundation of China (U23A20146 to L.T.).

## Author contributions

L.J.T. conceived the original idea; L.J.T., M.Y.Z. and X.J.P. designed research; M.Y.Z., X.J.P. and M.C. performed research; L.J.T., M.Y.Z., X.J.P. and M.C. analyzed data; L.J.T. and M.Y.Z. drafted the manuscript, and all authors revised the manuscript.

## Competing interests

The authors declare no competing interests.
