## [Peer Review File · Nature Communications]

REVIEWER COMMENTS

Reviewer #1 (Remarks to the Author):

The manuscript authored by Zheng and colleagues delves into a fascinating exploration of the physical mechanism and identity of the NPQ quencher in *Chlamydomonas* membranes. Notably, the authors conducted a pioneering study utilizing non-solubilized material—membrane fragments—derived from active NPQ cells of this alga. The reversible and substantial quenching observed in these fragments, coupled with their lower light scattering compared to cells, facilitated an impressive application of ultrafast absorption spectroscopy. The meticulous analysis uncovered the energy transfer path from chlorophyll to carotenoid in the NPQ state, with indications strongly pointing towards lutein as the likely carotenoid. Despite the anticipated low signal, the authors provided compelling evidence affirming the reality of the quencher signal, showcasing a state-of-the-art achievement. The overall work is convincing and well-presented. I would recommend expanding the last paragraph, incorporating reference [47], which provides crucial evidence for a non-Coulombic energy pathway from chlorophyll to carotenoid.

Reviewer #2 (Remarks to the Author):

In “Ultrafast energy quenching mechanism of LHCSR3-dependent photoprotection in *Chlamydomonas*”, Zheng, M. et. al. investigate the role of the key gene product, LHCSR3, in nonphotochemical quenching (NPQ) in green algae. Transient absorption measurements on cell fractions with different genotypes/pH conditions, which mimic different levels of quenching, were performed. Previous work on isolated LHCSR3 showed that the quencher is likely the Car S1 state (de la Cruz Valbuena, G. et. al., *J. Phys. Chem. Lett.*, 2019), and here the authors show that this is also true in cell fractions. The mechanism of quenching in the intact system is an important and open question in the field, and the approach has the potential to address this hole in the field. However, the level of quenching in the cell fractions is not quantified nor is the level of quenching in the transient absorption experiments. If this quantification was performed and shown to be consistent across samples, the work would be an important advance for the mechanism of quenching in the context of intact systems. Without such quantification and correlation, the work is more appropriate for a specialized journal.

Specific comments:

- The authors assert that quenching is maintained in cell fractions, and provide relative fluorescence spectra to illustrate this point. However, the axes are arbitrary, the protein content may be different, and so the level of quenching needs to be quantified through fluorescence yield/lifetime measurements.
- Page 5, Lines 147-148, and Figure 2, please include additional motivation for fitting the TA data into 5 states. Although the supplementary figures seem to show that the model fits the data well, the DADSs

extracted don't seem physical. Namely, the early time black and red DADS associated with chlorophyll seem to have nonphysical structure at the carotenoid edge and feature ESA signal at that Chl edge.

- Along with a better justified fit, the relative amplitude of the quencher DADSs needs to be reported to quantify the differential quenching in the samples. Alternatively, fit values for single wavelength kinetics at the Car S1 to SN transition could be reported.

- Page 8, Lines 258-260, to what extent is the lack of Car cation signature due to low probe counts in that region? How confidently can you rule out Chl-Car charge separation?

- Overall, the manuscript is not written in a manner accessible to a general audience. The language of the manuscript is difficult to parse due to a plethora of grammatical errors and run-on sentences. This is most prevalent in the abstract and introduction. I recommend a close proofread of these sections at the very least to eliminate issues. Here are some of the errors spotted in the abstract alone:

o Page 2, Lines 14-16 "This is because of the tremendous difficulty, if not impossible, in measuring intact cells with transient absorption techniques" seems ungrammatical

o Page 2, Line 17 "...that still function physiological qE" seems ungrammatical

o Page 2, Line 18 "Combining with kinetic modeling..." should be "Combined with kinetic modeling"

o Page 2, Line 20 improper use of the word "hence"

Some examples of run-on sentences include Page 2, Lines 41-44, Page 3, Lines 72-75, and Lines 77-79.

Minor comments:

- Page 4, Line 93, sst7-9 and npq4/sst7-9 are mentioned before being introduced later on in the following section

- Page 4, Line 113, the phrase "state transition" is unhyphenated here but hyphenated in subsequent lines 115-116

- Page 7, Line 225, there should be some indication as to what may be the cause of the absence of the Car-triplet signal in the raw data, even if the reason is not confirmed

- Page 8, line 238, there is no black line in figure 4B stated. In fact the red arrow seems to be pointing to a cyan line as the quencher (and seems otherwise indicated as such in Figure 4A)

- Page 16, Line 552, and Page 17, Lines 591-592, when writing the cuvette size, please clarify here which side/length is associated with the path length.

Reviewer #3 (Remarks to the Author):

I'm afraid this paper is not in a publishable form in its present state. It is poorly, and in parts, confusingly written and contains a number of incorrect statements. It also does not refer to earlier work, such as

Bonente et al., PLoS Biol (2011) where the effect of pH on LHCSR3 was described, including in transient absorption experiments.

The conclusion that their spectra do not provide evidence for carotenoid cation signals appears inconclusive because their transient spectra do not seem to extend beyond 900 nm, while carotenoid cation species frequently have spectra at longer wavelengths than 900 nm. In fact, the charge transfer state was previously detected in *N. oceanica* at 980 nm in Park et al., PNAS (2019) at 980 nm.

The last claim in the discussion section “the coulomb-mediated energy transfer to car ... could not explain the high rate of energy quenching since the S0 to S1 state transition in carotenoid is optically forbidden. Possibly, non-coulomb interaction in the Chl-car pair dominates the energy transfer process” is misleading at best, and confuses near-field and far-field coulomb interactions. Strong coulomb coupling between two molecules is perfectly possible, even though one of them has a forbidden optical transition.

The analysis of the transient absorption spectra is highly problematic. The authors claim that the first two components of the global analysis are only related to energy equilibration in LHCs, but the papers cited to support this claim are for isolated LHCII, not for the intact system. Additionally, there is clear dependence on pH for these components (Figure S11), so the claim that these timescales are unrelated to quenching is not substantiated. Also, the difference between figures S11C and S11H are quite small, but the authors claim that the pH effect in C is due to quenching but the effect in H is not.

The npq4/stt7-9 mutant, which the authors use as a control, is concerning because in Figure 1D, the peak fluorescence intensity is higher at pH 5.5 than pH 7.5.

The target analysis model is additionally problematic because the timescale associated with decay of the quencher (7.5 ps) is much faster than the timescale associated with the growth (100 ps), so it is not feasible to see the spectrum of the quencher in the SADS. This is a fundamental issue in the setup of the model.

We regret that we cannot have a more positive response to this paper.

Answers to the reviewers' comments point by point:

Reviewer #1 (Remarks to the Author):

The manuscript authored by Zheng and colleagues delves into a fascinating exploration of the physical mechanism and identity of the NPQ quencher in *Chlamydomonas* membranes. Notably, the authors conducted a pioneering study utilizing non-solubilized material—membrane fragments—derived from active NPQ cells of this alga. The reversible and substantial quenching observed in these fragments, coupled with their lower light scattering compared to cells, facilitated an impressive application of ultrafast absorption spectroscopy. The meticulous analysis uncovered the energy transfer path from chlorophyll to carotenoid in the NPQ state, with indications strongly pointing towards lutein as the likely carotenoid. Despite the anticipated low signal, the authors provided compelling evidence affirming the reality of the quencher signal, showcasing a state-of-the-art achievement. The overall work is convincing and well-presented. I would recommend expanding the last paragraph, incorporating reference [47], which provides crucial evidence for a non-Coulombic energy pathway from chlorophyll to carotenoid.

Response: We appreciate the positive comments and the suggestion of including ref [47] in the possible energy transfer mechanism discussion.

We have expanded the last paragraph, as shown below.

“The weak Coulomb-mediated Chl-car interaction when a dipole-dipole approximation was employed could not explain the high rate of energy quenching since the S_0 to S_1 state transition in the carotenoid is optically forbidden. Possibly, the strong Coulomb interaction and/or the exchange term in the Chl-car coupling dominates the energy transfer process^{87, 88}. This speculation agrees with the recent work of Ruan et al.⁴⁷, who assigned the energy transfer process from Chl to Lut to short-range (Dexter) energy transfer. They observed a sharp decline in lifetimes (from ns to a few hundred picoseconds per LHCII trimer) when the distance between Chl and Lut becomes closer than 5.6 Å in the purified LHCII in nanodiscs. This steep distance dependence could be fitted well with a Dexter-type EET model. However, it should be noted that compared to the LHCII in nanodiscs, the quenching process in a native membrane, as presented

here, is happening at least ten times faster (~100 ps per PSII complex).” See page 9, lines 281-291.

Reviewer #2 (Remarks to the Author):

In “Ultrafast energy quenching mechanism of LHCSR3-dependent photoprotection in *Chlamydomonas*”, Zheng, M. et. al. investigate the role of the key gene product, LHCSR3, in nonphotochemical quenching (NPQ) in green algae. Transient absorption measurements on cell fractions with different genotypes/pH conditions, which mimic different levels of quenching, were performed. Previous work on isolated LHCSR3 showed that the quencher is likely the Car S1 state (de la Cruz Valbuena, G. et. al., J. Phys. Chem. Lett., 2019), and here the authors show that this is also true in cell fractions. The mechanism of quenching in the intact system is an important and open question in the field, and the approach has the potential to address this hole in the field. However, the level of quenching in the cell fractions is not quantified nor is the level of quenching in the transient absorption experiments. If this quantification was performed and shown to be consistent across samples, the work would be an important advance for the mechanism of quenching in the context of intact systems. Without such quantification and correlation, the work is more appropriate for a specialized journal.

Response: We would like to thank the reviewer for these comments.

We agree that the quantification of energy quenching is critical. In the previous version, we have already quantified the quenching levels. From the steady-state fluorescence spectra that mainly originate from PSII, we found roughly 60% of the total fluorescence of *stt7-9* pH 5.5 sample was quenched when compared to the unquenched state at pH 7.5 (see page 5, lines 121-122). Accordingly, in our target model of the transient absorption data, we also estimated that roughly 60% of PSII was quenched during quenching, which led to a satisfactory fit (page 7, lines 227-228). As anticipated, the total fluorescence (reconstructed from the TA data in Figure 2C) of UQ and Q samples was not different by 60% but less, ~40% (Figure S14A), because of the presence of annihilation, even though the quenched fraction is consistent with the steady-state fluorescence data.

Figure S14. The steady-state fluorescence spectra were reconstructed from the DADSs of the two samples at different pHs (Figures 2C and F).

To explicitly emphasize this point, we have now added the following text to the manuscript:

“To quantify the quenching effect observed in the transient absorption data, we reconstructed the steady-state fluorescence based on DADSs, see Figure S14. The total fluorescence decreased by roughly 40% at lower pH for the *stt7-9* mutant while changing marginally in the double mutant of *npq4/stt7-9*. The quenching level is lower than that observed 60% in the steady-state measurements (Figure 1C). This discrepancy is caused by the unavoidable multi-exciton annihilation effect in ultra-short laser spectroscopy, which accelerates the overall decay of the system and thus diminishes the quenching.

To summarize, all the lifetime components, their possible assignments, and how they were affected by qE were summarized in Table S1. As expected, for double mutant, the overall quenching effect on the DADSs (Figure 2F) caused by LHCSR1 showed a similar pattern but much weaker as compared to the quenching in *stt7-9*, in which both LHCSR3 and LHCSR1 are functional.” See page 6, lines 178-188.

Specific comments:

- The authors assert that quenching is maintained in cell fractions, and provide relative fluorescence spectra to illustrate this point. However, the axes are arbitrary, the protein content may be different, and so the level of quenching needs to be quantified through fluorescence yield/lifetime measurements.

Response: We sincerely thank the reviewer for this comment. Indeed, we have normalized the absorption spectra at 676 nm to compare their shape. It becomes problematic in quantifying the absolute Δ -fluorescence between UQ and Q samples. At the beginning of the project, we checked and found no significant changes in absorption spectra before and after pH titration (8 μ L base per 2 mL solution). Therefore, we took it for granted that the absorption spectra would not change in amplitude. Unfortunately, the absorption spectra shown in Figure 1A and B in the previously submitted manuscript were collected with no attempt to compare their absolute values. Emission spectra remain valid because the pH titrations were made on-site. Namely, the cuvette (with magnetic stirring on) was kept in the measuring chamber throughout the measurement. To avoid misunderstanding, we replaced Figures 1C and D with raw data that were not normalized and modified the caption. For consistency, we have re-done all the measurements in Figure 1A, B and modified the figure and its caption accordingly, as shown below. Note that there are differences in the Soret band between the newly measured and the previously measured absorption spectra because the previous ones were measured with an integrating sphere, which is now out of order. The conclusion is not affected. See page 18.

Figure 1. Steady-state spectra of the micro-sized cell fractions of *C. reinhardtii* cells of *stt7-9* and *npq4/stt7-9*. (A) and (B) respectively represent the room temperature absorption spectra of *stt7-9* and *npq4/stt7-9* at both pH 5.5 and 7.5. Both were measured without using an integrating sphere. (C) and (D) respectively represent the room temperature fluorescence spectra of *stt7-9* and *npq4/stt7-9* at pH 5.5 and 7.5. Excitation was made at 430 nm.

- Page 5, Lines 147-148, and Figure 2, please include additional motivation for fitting the TA data into 5 states.

Response: We thank the reviewer for the suggestion. We have tried the parallel scheme with four, five, and six components, respectively, see the results below. The root mean square (RMS), usually the lower the better fit is, does become lower when we test model with more components. Taking the TA data of *stt7-9* at pH 5.5 as an example, the RMS of fit with a four-component model is 0.0371, with a five-component model 0.0364. and with a six-component model 0.0358. When the six-component model is used, the competitive effect between the second (red) and third (blue) DADS is getting significant, suggesting that five components are already the upper limit of the data that could resolve. Therefore, a model with five components has been presented.

We have added the following text in the manuscript to clarify this point.

“To quantitatively describe the TA data, we have employed a parallel kinetics model of 5 compartments, which is a compromise between simplicity and sufficient fit” See page 5, lines 146-148.

Figure: The DADS and lifetime of each component were calculated using a parallel scheme with four, five and six components, respectively.

Although the supplementary figures seem to show that the model fits the data well, the

DADSs extracted don't seem physical. Namely, the early time black and red DADS associated with chlorophyll seem to have nonphysical structure at the carotenoid edge and feature ESA signal at that Chl edge.

Response: We agree with the reviewer. In Figures S11F and G, DADSs of the two fastest components (solid lines) showed unexpected/nonphysical signals between 500 and 550 nm, this distortion is probably caused by a fitting artifact, namely, two DADS with comparable lifetimes compete with each other in the spectrum in the fitting, which leads to opposite amplitudes in a specific wavelength range even though the fit is still mathematically correct. To verify, we included the global analysis results on the TA data of another two biological repeats of the *npq4/stt7-9* samples. As shown below for review only, their DADSs are as expected and no significant changes were observed upon quenching.

However, we would like to continue to present the current fitting results since the S/N ratios of the presented series of datasets are the highest. The misfit in this region of the double mutant will not affect the main conclusion.

Figure. The DADSs of the TA data of another two biological repeats of the *npq4/stt7-9* samples. DADSs of the first two components are all shown in bold.

The rest of the DADSs are normal, DADS, different from EADS, are often sensitive to the lifetimes and spectral shape of donors and acceptors when EETs are involved as it is in this case.

- Along with a better justified fit, the relative amplitude of the quencher DADSs needs to be reported to quantify the differential quenching in the samples. Alternatively, fit

values for single wavelength kinetics at the Car S1 to SN transition could be reported.

Response: We thank the reviewer for this suggestion. The decay trace of *stt7-9* pH 5.5 sample at 540 nm and the population of different compartments at the single wavelength are shown below. We have changed Figure 4A and modified the caption in the main text accordingly. See page 21, lines 653-655.

Figure 4A. The kinetics trace of the *stt7-9* pH 5.5 sample at 540 nm and the populations of different compartments at this wavelength were estimated in the global target analysis (data of the *stt7-9* sample at pH 5.5 upon 675 nm excitation).

- Page 8, Lines 258-260, to what extent is the lack of Car cation signature due to low probe counts in that region? How confidently can you rule out Chl-Car charge separation?

Response: We thank the reviewer for these critical questions. We have now improved the S/N ratio of the TA data in the NIR region and extended the detection wavelength window down to 1000 nm, and our conclusion remains the same that no Car cation absorption observed, see replaced Figure S9 in the Supporting Information and below. According to the published work on Car cations (Alberta Pinnola et al., *Biochimica et Biophysica Acta*, 2016; Gabriel de la Cruz Valbuena et al., *J. Phys. Chem. Lett.*, 2019), the maximum signal in the NIR region accounts up to ~10% of the bleaching signal at 680 nm, which is less likely overlooked in our experiments if it is there. More importantly, there is no difference in this region for the sample at different pHs. Therefore, the Chl-carotenoid charge separation model can be safely ruled out.

Figure S9. TA data of *stt7-9* micro-sized cell fractions in 620-1000 nm region. (A) The TA spectra at three different time-points of *stt7-9* samples at pH 5.5 and pH 7.5. Inset: The zoom-in spectra in the near-infrared region (800-1000 nm). (B) TA kinetics at 850, 900, and 950 nm of *stt7-9* samples at different pHs. The spectra and traces at different pHs were normalized to their bleaching maximum at around 680 nm at time zero.

- Overall, the manuscript is not written in a manner accessible to a general audience. The language of the manuscript is difficult to parse due to a plethora of grammatical errors and run-on sentences. This is most prevalent in the abstract and introduction. I recommend a close proofread of these sections at the very least to eliminate issues. Here are some of the errors spotted in the abstract alone:

Response: Thank the reviewer for being patient with us on the language issue. We have corrected the errors and carefully improved the text. The modifications are shown with track changes.

o Page 2, Lines 14-16 “This is because of the tremendous difficulty, if not impossible, in measuring intact cells with transient absorption techniques” seems ungrammatical
Corrected. See page 2, lines 14-16.

o Page 2, Line 17 “...that still function physiological qE” seems ungrammatical
Corrected. See page 2, lines 16-18.

o Page 2, Line 18 “Combining with kinetic modeling...” should be “Combined with kinetic modeling”
Corrected. See page 2, line 18.

o Page 2, Line 20 improper use of the word “hence”

Corrected. See page 2, line 20.

Some examples of run-on sentences include Page 2, Lines 41-44, Page 3, Lines 72-75, and Lines 77-79.

Modified, see page 2, lines 41-44, page 3, lines 71-75, page 3, lines 77-79.

Minor comments:

- Page 4, Line 93, *sst7-9* and *npq4/sst7-9* are mentioned before being introduced later on in the following section

Response: Thanks. We have shifted the introduction of *sst7-9* and *npq4/sst7-9* to page 4, lines 85-89.

- Page 4, Line 113, the phrase “state transition” is unhyphenated here but hyphenated in subsequent lines 115-116

Modified. See page 4, line 88.

- Page 7, Line 225, there should be some indication as to what may be the cause of the absence of the Car-triplet signal in the raw data, even if the reason is not confirmed

Response: Thanks for noticing this detail. Our educated guess would be that the effective pump pulse density on PSII is lower for the double mutant as it contains less PSII (40% of the total Chls) than the *sst7-9* mutant does (70% of the total Chls), i.e., PSII is more shaded by PSI in the double mutant.

We have clarified this point in the main text, “The absence of a Car-triplet in the raw data may be because the effective exciting intensity on PSII is lower since the mutant contains higher proportion PSI (~60%), which could shade PSII, thus producing less triplet state.” See page 8, lines 237-239.

- Page 8, line 238, there is no black line in figure 4B stated. In fact the red arrow seems to be pointing to a cyan line as the quencher (and seems otherwise indicated as such in Figure 4A)

Response: Corrected. We sincerely apologize for this mistake. See page 8, line 251 and 257, page 21, line 655.

- Page 16, Line 552, and Page 17, Lines 591-592, when writing the cuvette size, please clarify here which side/length is associated with the path length.

Added, see page 15, line 543 and page 16, line 582-583.

Reviewer #3 (Remarks to the Author):

I'm afraid this paper is not in a publishable form in its present state. It is poorly, and in parts, confusingly written and contains a number of incorrect statements.

Response: We sincerely thank the reviewer for pointing out the problems in writing and incorrect statements. It certainly helps to improve the quality of the manuscript once all the problems are fixed. We have carefully revised the text and addressed all the concerns raised by the reviewer point by point, see below. With these improvements, we hope that the manuscript is now in a publishable form.

It also does not refer to earlier work, such as Bonente et al., PLoS Biol (2011) where the effect of pH on LHCSR3 was described, including in transient absorption experiments.

Response: We fully agree with the reviewer that the earlier work (Bonente G et al., PLOS Biol, 2011) is highly relevant. Therefore, we have cited this reference three times, ref. 26. See page 3, line 54, page 4, line 106, and page 9, line 274. The statement “It also does not refer to earlier work” is incorrect.

The conclusion that their spectra do not provide evidence for carotenoid cation signals appears inconclusive because their transient spectra do not seem to extend beyond 900 nm, while carotenoid cation species frequently have spectra at longer wavelengths than 900 nm. In fact, the charge transfer state was previously detected in *N. oceanica* at 980 nm in Park et al., PNAS (2019) at 980 nm.

Response: We thank the reviewer for this suggestion. Indeed, as pointed out by the reviewer, carotenoid cation signals also appear beyond 900 nm in some cases. Therefore, we have again measured the TA data in the NIR region with a much-improved S/N ratio and a broader detection window up to 1000 nm, see figure below and updated Figure S9 in the Supporting Information. Still, we failed to pinpoint any positive peak in the 800-1000 nm region and spectral differences for the sample at different pHs. Thus, the Chl-carotenoid charge separation model can be excluded for this specific system.

Figure S9. TA data of *stt7-9* micro-sized cell fractions in 620-1000 nm region. (A) The TA spectra at three different time-points of *stt7-9* samples at pH 5.5 and pH 7.5. Inset: The zoom-in spectra in the near-infrared region (800-1000 nm). (B) TA kinetics at 850, 900, and 950 nm of *stt7-9* samples at different pHs. The spectra and traces at different pHs were normalized to their bleaching maximum at around 680 nm at time zero.

The last claim in the discussion section “the coulomb-mediated energy transfer to car ... could not explain the high rate of energy quenching since the S₀ to S₁ state transition in carotenoid is optically forbidden. Possibly, non-coulomb interaction in the Chl-car pair dominates the energy transfer process” is misleading at best, and confuses near-field and far-field coulomb interactions. Strong coulomb coupling between two molecules is perfectly possible, even though one of them has a forbidden optical transition.

Response: We sincerely thank the reviewer for pointing out the inaccurate statement. We have modified this part of the discussion, see the text below and in the main text.

“The weak Coulomb-mediated Chl-car interaction when a dipole-dipole approximation was employed could not explain the high rate of energy quenching since the S₀ to S₁ state transition in the carotenoid is optically forbidden. Possibly, the strong Coulomb interaction and/or the exchange term in the Chl-car coupling dominates the energy transfer process^{87, 88}. This speculation agrees with the recent work of Ruan et al.⁴⁷, who assigned the energy transfer process from Chl to Lut to short-range (Dexter) energy transfer. They observed a sharp decline in lifetimes (from ns to a few hundred

picoseconds per LHCII trimer) when the distance between Chl and Lut becomes closer than 5.6 Å in the purified LHCII in nanodiscs. This steep distance dependence could be fitted well with a Dexter-type EET model. However, it should be noted that compared to the LHCII in nanodiscs, the quenching process in a native membrane, as presented here, is happening at least ten times faster (~100 ps per PSII complex).” See page 9, lines 281-291.

The analysis of the transient absorption spectra is highly problematic. The authors claim that the first two components of the global analysis are only related to energy equilibration in LHCs, but the papers cited to support this claim are for isolated LHCII, not for the intact system. Additionally, there is clear dependence on pH for these components (Figure S11), so the claim that these timescales are unrelated to quenching is not substantiated.

Response: We thank the reviewer for this comment. Indeed, the papers cited here are on isolated LHCII/aggregates and CP29. First of all, LHCII are by far the dominant proteins in LHCs. Furthermore, energy equilibrium within LHCII or between LHCII is likely realized in the same manner as in its native membrane. Therefore, it is highly relevant to cite these papers. Secondly, to our knowledge, no literature yet reports detailed femtosecond kinetics of PSII-LHCII supercomplexes at this Chlorophyll bleaching region. Thus, we have only cited papers on LHCs.

We agree that the first two components in Figure S11 are pH-dependent around the Chl bleaching signal. We apologize for overlooking these differences. The bleaching signal at around 680 nm of *stt7-9* pH 5.5 sample is higher than at pH 7.5, which might be induced by an accelerated energy equilibrium between different Chl pools when quenched. We have revised this sentence in the manuscript and Table S1 in the Supporting Information.

"Energy equilibrium between different pools of chlorophylls (black and red DADSs in Figure 2C) takes place in 300 fs and 1.8 ps. These two processes were previously observed for the purified LHCs/aggregates, where excitation reaches equilibrium within sub-picosecond to a few picoseconds^{42, 43, 65-67}. Note that there might be slower EET processes with lifetimes of around ten picoseconds in the system. However, our

model does not resolve these because of its low population (due to the nearly complete spectral overlap between donor and acceptor molecules). Upon quenching, these two fast processes were both enhanced at 680 nm for the mutant of *stt7-9* (Figures S11A and B), but neither was changed in the double mutant of *npq4/stt7-9* (Figures S11F and G). These results indicate that the LHCSR-dependent quenching processes are fast enough to disturb this equilibrium." See page 5, lines 152-161.

Also, the difference between figures S11C and S11H are quite small, but the authors claim that the pH effect in C is due to quenching but the effect in H is not.

Response: We did not make such a claim. Instead, we concluded that in the double mutant sample, the quenching effect on DADSs is weaker because only LHCSR1 is present, see page 6, lines 186-188.

The *npq4/stt7-9* mutant, which the authors use as a control, is concerning because in Figure 1D, the peak fluorescence intensity is higher at pH 5.5 than pH 7.5.

Response: Thank you for noticing this detail. We speculate that the fluorescence intensity is lower at pH 7.5 than 5.5 in Figure 1D because a small amount of the sample is damaged by the transient and locally high pH at the pipette tip during KOH addition. We consider it an acceptable measurement error.

The target analysis model is additionally problematic because the timescale associated with decay of the quencher (7.5 ps) is much faster than the timescale associated with the growth (100 ps), so it is not feasible to see the spectrum of the quencher in the SADS. This is a fundamental issue in the setup of the model.

Response: As the reviewer pointed out, we are investigating a typical "inverted kinetic model" (see page 8, lines 224-227). This "inverted kinetic model" has been detected in several different LHC protein systems, and the spectrum of the quencher was also obtained by target analysis of the TA data (Alexander V. Ruban et al., Nature, 2007; Vincenzo Mascoli., Chem, 2019; Nicoletta Liguori et al., Nature Communications, 2017; Mengyuan Zheng et al., J. Phys. Chem. Lett, 2023), even though the population is small (see updated Figure 4A). As we have proven here, resolving the spectrum of the quencher is feasible.

We regret that we cannot have a more positive response to this paper.

Response: Again, we sincerely thank the reviewer for these constructive comments, and we have carefully responded to the comments and modified the text accordingly. We hope the manuscript can now be considered publishable.

REVIEWER COMMENTS

Reviewer #1 (Remarks to the Author):

The authors clearly responded to my comments.

Reviewer #2 (Remarks to the Author):

We thank the Zheng, et. al. for their response, particularly in providing additional data comparing steady-state fluorescence and reconstructed DADS from global analysis of the transient absorption data. Although this does show that transient absorption data and the fluorescence data are consistent with each other, it doesn't show the contribution of each individual DADS. Quantification of the amplitude of the quencher in the DADS should be included. Table S1 discusses the amplitude of DADS contributions, but only by comparing the amplitude change between sst7-9 and npq4/sst7-9 samples at pH 5.5, and thus does not provide the requested quantification. With only comparative amplitudes between samples, the contributions of different DADs for each sample cannot be compared. Explicit amplitude values in both samples should also be reported in this table.

More importantly, the amplitudes of each state present in the TA data as assigned in the global target scheme in Figure 3(A) should be stated explicitly. Currently, we can only make a qualitative assessment of contributions from the SADS states at 540 nm based on Figure 4(A). Reporting amplitudes of SADS states from the target model explicitly will help determine the contribution of the "Quencher" state, i.e., how much energy is dissipated through Lut1 (as opposed to annihilation quenching, for example). If the authors are willing to provide quantitative rather than simply qualitative values for the DADS (assuming the "Quencher" contribution is significant, which cannot be evaluated in the current form), we believe that the manuscript will be fit for publication.

Reviewer #3 (Remarks to the Author):

The revised manuscript by Zheng, M. et al., is significantly improved in the data interpretation and has clarified previously ambiguous evidence. The paper is in a publishable form, while some minor edits are recommended to enhance clarity and informativeness further.

Comments

1.

Fig 3

The kinetic model and SADS suggest an equilibrium between two Chl pools after the initial excitation. If feasible, assigning each Chl pool to possible sites in the membrane system would provide additional information. Moreover, the SADS of Chl2 and PSII/RP show remarkable similarity for both the stt7-9 and double mutant. Although this assumption is contradicted to current model, could PSII/RP be a member of the Chl2 pool?

2. The text in S1 describes the constraints for analyzing SADS, where the authors use the product of the excited PSII ratio at pH 7.5 (65%) and fluorescence reduction ratio (60%) to represent the excitation ratio of the PSII_Q pathway at pH 5.5. However, it is important to note that, in the PSII_Q pathway, only a portion of the excitation is quenched by the quencher and leads to fluorescence reduction. Therefore, the evaluated quenched ratio (60%*65%) should account for the population that “reaches quencher” rather than “undergoes PSII_Q pathway”. The original evaluation may underestimate the actual population of PSII_Q pathway.

3.

Fig S9

The investigation of Chl-Carotenoid charge transfer kinetics, over an expanded spectral range up to 1000 nm along with individual kinetic profiles analysis, provides evidence that the transient population of charge separation is not resolvable. This clarified the negligible possibility of influence from charge transfer kinetics in this experiment.

Fig S11 (A-B) / Lines 152-161

The authors have expanded their interpretation to elucidate the differences observed in the initial DADSs (300 fs and 1.8 ps) between pH 5.5 and pH 7.5. This further analysis has provided clarification regarding the influence of LHCSR-dependent quenching on the earlier time equilibration.

4.

Lines 283-284

The authors interpret the rapid Chl to carotenoid energy transfer rate ($>1 \text{ ps}^{-1}$) as being attributed to “strong Coulomb interaction and/or the exchange term in the Chl-car coupling dominates the energy transfer process”. These concepts are related to the Forster- and Dexter-type excitonic coupling discussed in Madjet, M. et. al. *J. Phys. Chem. B* (2009). The cited references in the revision also includes this paper and provides the distance between Chl and Lut in LHCI. However, I think the author may be overly focused on Dexter-type coupling and may have overlooked the potential contribution of Forster-type coupling in their interpretation. The authors may wish to look at *J Phys Chem B* 105, 11016 (2001) and *J Phys Chem A* 106, 1909 (2002).

5.

Lines 224-227 / Fig 4

The authors explain the kinetics of the quencher, characterized by a slow rise (100 ps) and fast decay (7.5 ps), through an "inverted kinetic model" and demonstrate that the transient population of the quencher is resolvable. Given a sufficiently large population of excited PSII/RP, the production rate would be high, leading to the accumulation of the quencher population. Thus, I find the authors' interpretation to be reasonable.

Reviewer #1 (Remarks to the Author):

The authors clearly responded to my comments.

Reviewer #2 (Remarks to the Author):

We thank the Zheng, et. al. for their response, particularly in providing additional data comparing steady-state fluorescence and reconstructed DADS from global analysis of the transient absorption data. Although this does show that transient absorption data and the fluorescence data are consistent with each other, it doesn't show the contribution of each individual DADS. Quantification of the amplitude of the quencher in the DADS should be included. Table S1 discusses the amplitude of DADS contributions, but only by comparing the amplitude change between *stt7-9* and *npq4/stt7-9* samples at pH 5.5, and thus does not provide the requested quantification. With only comparative amplitudes between samples, the contributions of different DADS for each sample cannot be compared. Explicit amplitude values in both samples should also be reported in this table.

Response: We thank the reviewer for this suggestion. To show the contributions of each DADS to the reconstructed steady-state fluorescence, we modified Figure S14 below.

Figure S14. The steady-state fluorescence spectra were reconstructed from the DADSs of the two samples at different pHs (Figures 2C and F). The wine-dotted and solid lines represent the sum of the reconstructed spectra of each lifetime component of the sample at pH 5.5 and 7.5, respectively.

The last column in Table S1 shows the amplitude changes of DADSs at pH 5.5 and 7.5 for the *stt7-9* and *npq4/stt7-9* samples, but not between *stt7-9* and *npq4/stt7-9* samples

at pH 5.5. We should have made it more clear in the previous version. To show the changes in DADSs at different pHs, we have modified Table S1 below, and accordingly, the description in the manuscript. See page 6, lines 168, 174-175, and 189-190.

Table S1. The lifetime, assignment of the DADSs for *stt7-9* (Figure 2C), and the calculated proportion of the Chl-Q_y bleaching at 680 nm to the total at this wavelength.

***stt7-9* pH 7.5/*stt7-9* pH 5.5**

lifetime	assignment	The proportion of each DADS at 680 nm to the total Chl-Q _y bleaching
55 ps/54 ps	PSI, annihilation, and PSII quenching	37%/48%
250 ps/280 ps	PSII quenching and PSII_CS	25%/29%
1.2 ns	PSII_CS	38%/23%

EET: Excitation Energy Transfer; CS: Charge Separation

More importantly, the amplitudes of each state present in the TA data as assigned in the global target scheme in Figure 3(A) should be stated explicitly. Currently, we can only make a qualitative assessment of contributions from the SADS states at 540 nm based on Figure 4(A). Reporting amplitudes of SADS states from the target model explicitly will help determine the contribution of the “Quencher” state, i.e., how much energy is dissipated through Lut1 (as opposed to annihilation quenching, for example). If the authors are willing to provide quantitative rather than simply qualitative values for the DADS (assuming the “Quencher” contribution is significant, which cannot be evaluated in the current form), we believe that the manuscript will be fit for publication.

Response: We thank the reviewer for this suggestion. We have now added to Figure 4 the concentrations of the different species estimated in the global target analysis, and we highlighted the differences between the populations of PSII and RP caused by quenching, see Figure 4A.

Figure 4. Proposed energy quenching mechanism of LHCSR3-related qE in the native thylakoid membrane. (A) Concentrations of the different species were estimated in the global target analysis (data of the *stt7-9* sample at pH 5.5 upon 675 nm excitation). For comparison, the solid curves are 1.5-fold magnified, and the areas in magenta and green represent the population changes of PSII and RP, respectively, due to quenching. For simplicity, the concentrations of two branches, PSI and Car-Triplet, are omitted. (B) The kinetics trace of the *stt7-9* pH 5.5 sample at 540 nm and the populations of different compartments at this wavelength. (C) The spectrum of the quencher (solid cyan line) that overlaid with the ESA spectrum of carotenoid S₁ (blue dashed line, see details in Figure S19B) from TA data of *stt7-9* micro-sized cell fractions at pH 5.5 and several published ESA spectra of Lut1-S₁ state of LHCII aggregates (solid gray line)⁴³, carotenoid S₁ state of recombinant LHCSR3³⁸ (magenta dashed line) and carotenoid S₁ state of LHCII⁷⁶ (green dashed line). (D) Schematic of PSI and PSII super-complexes together with LHCSR3 in the thylakoid membrane.

Reviewer #3 (Remarks to the Author):

The revised manuscript by Zheng, M. et al., is significantly improved in the data interpretation and has clarified previously ambiguous evidence. The paper is in a publishable form, while some minor edits are recommended to enhance clarity and informativeness further.

We thank the reviewer for the positive comments.

Comments

1. Fig 3

The kinetic model and SADS suggest an equilibrium between two Chl pools after the initial excitation. If feasible, assigning each Chl pool to possible sites in the membrane system would provide additional information.

Response: We thank the reviewer for the suggestion. Unfortunately, based on the available information, we could not assign each Chl pool to its possible sites.

Moreover, the SADS of Chl2 and PSII/RP show remarkable similarity for both the stt7-9 and double mutant. Although this assumption is contradicted to current model, could PSII/RP be a member of the Chl2 pool?

Response: In the target analysis, we have simplified the target model as much as possible meanwhile ensuring high-quality fitting. Further simplification from the current model (grouping PSII/RP into Chl 2 pool) resulted in a misfit. We agree with the reviewer that if the pool of Chl 2 belongs to the periphery antenna, it should be slightly blue-shifted than the core antenna, e.g., when Chl b was selectively excited. However, this is different when excited at 675 nm. This is also the reason why we did not assign Chl 1 and Chl 2 specifically, see above.

2. The text in S1 describes the constraints for analyzing SADS, where the authors use the product of the excited PSII ratio at pH 7.5 (65%) and fluorescence reduction ratio (60%) to represent the excitation ratio of the PSII_Q pathway at pH 5.5. However, it is important to note that, in the PSII_Q pathway, only a portion of the excitation is quenched by the quencher and leads to fluorescence reduction. Therefore, the evaluated quenched ratio (60%*65%) should account for the population that “reaches quencher” rather than “undergoes PSII_Q pathway”. The original evaluation may underestimate

the actual population of PSII_Q pathway.

Response: We thank the reviewer for the comment. It should be noted that we estimated the quenching fraction of PSII to be roughly 60% for *stt7-9* sample at pH 5.5 based on the steady-state fluorescence data, and essentially, there is no annihilation with lamp light excitation. However, upon femtosecond laser excitation, annihilation is inevitable. Although annihilation won't vary the quenching fraction of PSII, it diminishes the overall quenching. As expected, the reconstructed steady-state bleaching signal from the TA data only shows ~40% quenching, see Figure S14. If annihilation were absent, the quenching fraction $60% \times 65%$ would remain the same; only the net quenching effect is more prominent.

Another point that might be overlooked is that the PSII fluorescence dominates, whereas fluorescence from the PSI branch is neglectable because of its fast kinetics.

3. Fig S9

The investigation of Chl-Carotenoid charge transfer kinetics, over an expanded spectral range up to 1000 nm along with individual kinetic profiles analysis, provides evidence that the transient population of charge separation is not resolvable. This clarified the negligible possibility of influence from charge transfer kinetics in this experiment.

Fig S11 (A-B) / Lines 152-161

The authors have expanded their interpretation to elucidate the differences observed in the initial DADSs (300 fs and 1.8 ps) between pH 5.5 and pH 7.5. This further analysis has provided clarification regarding the influence of LHCSR-dependent quenching on the earlier time equilibration.

4. Lines 283-284

The authors interpret the rapid Chl to carotenoid energy transfer rate ($>1 \text{ ps}^{-1}$) as being attributed to “strong Coulomb interaction and/or the exchange term in the Chl-car coupling dominates the energy transfer process”. These concepts are related to the Forster- and Dexter-type excitonic coupling discussed in Madjet, M. et. al. J. Phys. Chem. B (2009). The cited references in the revision also includes this paper and provides the distance between Chl and Lut in LHCI. However, I think the author may

be overly focused on Dexter-type coupling and may have overlooked the potential contribution of Forster-type coupling in their interpretation. The authors may wish to look at J Phys Chem B 105, 11016 (2001) and J Phys Chem A 106, 1909 (2002).

Response: We agree with the reviewer that the mixing of the S_2 state in Coulomb coupling induced by symmetry-breaking of the carotenoid might play an important role in the EET coupling strengths between Car S_1 and other pigments (J Phys Chem B 105, 11016, 2001). Thus, we have modified the manuscript as follows.

“The weak Coulomb-mediated Chl-car interaction when a dipole-dipole approximation was employed could not explain the high rate of energy quenching since the S_0 to S_1 state transition in the carotenoid is optically forbidden unless the S_1 state carries part of the S_2 oscillator strength due to symmetry-breaking of the carotenoid, as reported previously⁸⁷. Alternatively, the strong Coulomb interaction and/or the exchange term in the Chl-car coupling dominates the energy transfer process^{88,89}. This speculation agrees with the recent work of Ruan et al.⁴⁷, who assigned the energy transfer process from Chl to Lut to short-range (Dexter) energy transfer. They observed a sharp decline in lifetimes (from ns to a few hundred picoseconds per LHCII trimer) when the distance between Chl and Lut becomes closer than 5.6 Å in the purified LHCII in nanodiscs. This steep distance dependence could be fitted well with a Dexter-type EET model. Notably, compared to the LHCII in nanodiscs, the quenching process in a native membrane, as presented here, is happening at least ten times faster (~100 ps per PSII complex).” See page 9, lines 288-299.

Regarding the involvement of a hot Car S_1 state in quenching, we need more evidence to support a detailed discussion on this point. Here, we would like to keep the discussion simple.

5. Lines 224-227 / Fig 4

The authors explain the kinetics of the quencher, characterized by a slow rise (100 ps) and fast decay (7.5 ps), through an "inverted kinetic model" and demonstrate that the transient population of the quencher is resolvable. Given a sufficiently large population of excited PSII/RP, the production rate would be high, leading to the accumulation of the quencher population. Thus, I find the authors' interpretation to be reasonable.

REVIEWERS' COMMENTS

Reviewer #2 (Remarks to the Author):

The authors have satisfactorily addressed my comments and I now recommend the manuscript for publication.